# Joint Application of *Lactobacillus plantarum* and *Bacillus subtilis* Improves Growth Performance, Immune Function and Intestinal Integrity in Weaned Piglets

**DOI:** 10.3390/vetsci9120668

**Published:** 2022-11-30

**Authors:** Yisi Liu, Wei Gu, Xiaoyi Liu, Youwei Zou, Yujun Wu, Youhan Xu, Dandan Han, Junjun Wang, Jinbiao Zhao

**Affiliations:** 1State Key Laboratory of Animal Nutrition, College of Animal Science and Technology, China Agricultural University, Beijing 100193, China; 2Shandong Provincial Key Laboratory of Animal Microecological Agent, Shandong Baolai Leelai Bioengineering Co., Ltd., Tai’an 271000, China

**Keywords:** probiotics, weaned piglets, immune, intestinal health, gut microbiota

## Abstract

**Simple Summary:**

Problems such as the emergence of drug-resistant genes and food safety are becoming increasingly prominent due to the overuse of antibiotics. Probiotics can suppress the growth of pathogens effectively and enhance the immune functions of the host to replace antibiotics partially without the production of drug-resistance genes. There is an urgent request to explore the potential mechanisms of probiotics in regulating host health. In the study, our findings indicated that joint application of *Lactobacillus plantarum* and *Bacillus subtilis* as an antibiotics alternative improved growth performance via promoting immune function and intestinal integrity of weaned piglets.

**Abstract:**

This study was conducted to explore the effects of the joint application of *Lactobacillus plantarum* and *Bacillus subtilis* on growth performance, immune function, antioxidant capacity, intestinal integrity, and gut microbiota composition in weaned piglets. The piglets were allocated randomly into 4 dietary groups, which were a control diet (NC), NC + 150 ppm mucilage sulfate (PC), and 3 additional diets containing 1 kg/t (LT), 1.5 kg/t (MT), or 2 kg/t (HT) mixture of *Lactobacillus plantarum* and *Bacillus subtilis*, respectively. Results showed that joint application of *Lactobacillus plantarum* and *Bacillus subtilis* increased ADFI and ADG of weaned piglets in d 14~28 and d 28~42 (*p* < 0.05), and decreased serum concentrations of DAO, IL-1β, TNF-α, and IL-2. The LT group increased jejunal and colonic sIgA contents compared with the PC group (*p* < 0.05). Groups of MT and HT increased colonic mRNA expression of host defense peptides and tight junction proteins compared with the NC and PC groups. The joint application of *Lactobacillus plantarum* and *Bacillus subtilis* increased the abundance of colonic *Lactobacillus* compared with NC and PC groups (*p* < 0.10). In conclusion, the joint application of *Lactobacillus plantarum* and *Bacillus subtilis* as an antibiotics alternative improved growth performance via promoting immune function and intestinal integrity of weaned piglets.

## 1. Introduction

The early weaning of piglets is a crucial technology in the process of intensive pig production [1]. However, it always leads to an imbalance of intestinal microecology and damage of gut integrity due to oxidation stress, resulting in low feed intake and more diarrhea incidence in piglets [2]. Antibiotics have always been used in the past decades as growth promoters to relieve oxidation stress and cure diarrhea of piglet [2,3]. However, the abuse of antibiotics causes the propagation of drug-resistant pathogens and antibiotic residues in animal products [4]. Therefore, a worldwide prohibition on the abuse of antibiotics in animal husbandry has been carried out [5].

Probiotics are one of the most common antibiotics alternatives in animal husbandry and play a positive role in maintaining the integrity of the intestinal barrier, suppressing the growth of pathogenic bacteria, reducing piglets’ diarrhea and improving growth performance [6,7]. *Lactobacillus* supplemented into the diets of weaned piglets can relieve oxidation stress and improve pig performance by optimizing the composition of intestinal microbiota [8]. *Lactobacillus plantarum* improved the expression of host defense peptides in the pig intestine by regulating the TLR2/MAPK/AP-1 signaling pathway, resulting in improved immune function and growth performance [9]. In addition, *Bacillus* improves the intestinal health of pigs by modulating immune function and stimulating the growth of beneficial bacteria [10]. Supplementation of *Bacillus subtilis* increased the relative number of Lactobacillus and decreased the diarrhea incidence of piglets [11]. In addition, *Bacillus subtilis* supplementation improved the intestinal barrier function of intrauterine growth retardation piglets [12], as well as enhanced the abundance and structure of jejunal microorganisms [13]. However, it is difficult to achieve the goal of replacing antibiotics in the diets of weaned piglets with one strain of probiotics alone.

We hypothesized that joint application of *Lactobacillus plantarum* and *Bacillus subtilis* could be used as an antibiotic alternative to improve immune function and intestinal health in weaned piglets, resulting in better growth performance. Therefore, the purpose of this study was to evaluate the effects of joint application of *Lactobacillus plantarum* and *Bacillus subtilis* on growth performance, intestinal integrity, antioxidant capacity, immune function and gut microbiota community in weaning piglets.

## 2. Materials and Methods

Animal protocols were conducted based on regulations of the China Agricultural University Animal Care and Use Committee (Beijing, China) (AW92602202-1-1). The study was performed in the Feng Ning Swine Research Unit of China Agricultural University (Chengdejiuyun Agricultural and Livestock Co., Ltd., Hebei, China). Previous data in practice indicate that it is when numbers of 1.0 × 10^8^ CFU/g *Lactobacillus plantarum* or 1 × 10^9^ CFU/g *Bacillus subtilis* could improve weaned piglets’ performance. Then, a mixture of *Lactobacillus plantarum* and *Bacillus subtilis* were cultured in vitro fermentation. After optimizing culture conditions, the mixture of 2.0 × 10^8^ CFU/g *Lactobacillus plantarum* and 1.3 × 10^9^ CFU/g *Bacillus subtilis* was produced and used in our study.

### 2.1. Animals, Diets, and Experimental Design

Two hundred forty weaned piglets (Duroc × Landrace × Yorkshire) with an initial body weight of 7.9 ± 0.3 kg were randomly divided into five groups based on a litter of origin and gender. Piglets from the same litter were divided into dietary groups. There were six replicate pens for each dietary group, with eight piglets per pen. (4 gilts and 4 boars). Dietary treatments included a control diet (NC), NC + 150 ppm mucilage sulfate (PC) and 3 additional diets containing 1 kg/t (LT), 1.5 kg/t (MT), 2 kg/t (HT) mixture of *Lactobacillus plantarum* (2.0 × 10^8^ CFU/g) and *Bacillus subtilis* (1.3 × 10^9^ CFU/g). The positive control diet was chosen to use mucilage sulfate, an antibiotic used in the past for growth promotion in animal husbandry, in order to compare whether the addition of probiotics could also have a growth-promoting effect as an alternative to antibiotics. The ingredients composition of the NC diet and nutritive levels are shown in Table 1. The feeding trial lasted 42 days. All diets were supplemented with vitamin and trace minerals premixes to meet the nutrient requirement of weaned piglets. Piglets were fed in a nursery house with dung floors made of plastic leakage. Throughout the experiment, water and feed were freely available.

### 2.2. Sample Collection

In the feeding trial, all piglets were weighed individually on days 14, 28, and 42 to calculate average daily gain (ADG). The piglets’ feed consumption in each pen was recorded from d 1–14, d 14–28 and d 28–42 to calculate the average daily feed intake (ADFI). Feed conversion ratio (FRC) was defined as a ratio of ADFI to ADG. Furthermore, the diarrhea incidence was calculated using the following equation.

Diarrhea incidence (%) = 100 * [total number of diarrhea cases in each pen during the trial period/(total number of pigs in each pen * total number of days in the trial)].

On day 28, one piglet from each pen close to the average BW of the pen was selected to collect 10 mL blood sample using the anterior vena cava, and then centrifuged at 3000 r/min and 4 °C for 15 min to collect the serum and then stored at −20 °C for the further analysis. And the piglet was executed after anesthesia to collect samples of intestinal tissue, mucosa and digesta. To limit the variability of the samples, intestinal segments were collected from the middle part of the pig intestine (jejunum and colon). The jejunum and colon lumens were flushed with saline and fixed in a 4% para-formaldehyde solution for histological analysis. The intestinal epithelium was separated from the muscle layer using a blunt instrument and stored at −80 °C before being analyzed using ELISA and intestinal mucosal mRNA expression. Furthermore, before microbial genomic DNA extraction, intestinal chyme was immediately frozen in liquid nitrogen and stored at −80 °C.

### 2.3. Analysis of Samples

Serum concentrations of D-lactate and diamine oxidase (DAO) were analyzed by ELISA Kit (BMASSAY, China). Serum concentrations of aspartate aminotransferase (AST), glutamic alanine aminotransferase (ALT), and alkaline phosphatase (ALP) were analyzed by Micro Assay Kit (Solarbio Biotechnology Co., Ltd., Beijing, China). The serum concentration of Immunoglobulin G (IgG), Immunoglobulin M (IgM), and Immunoglobulin A (IgA) were analyzed by ELISA Kit (BMASSAY Biotechnology Co., Ltd., Beijing, China). Serum and intestinal concentrations of total antioxidant capacity (T-AOC) and superoxide dismutase (SOD) were analyzed using by activity assay kit (Solarbio Biotechnology Co., Ltd., Beijing, China). Serum and intestinal concentrations of malondialdehyde (MDA) were analyzed using by MDA assay kit (TBA method) (Jiancheng Institute of Bioengineering, Nanjing, China). The levels of interleukin-1β (IL-1β), interleukin-2 (IL-2), interleukin-10 (IL-10), tumor necrosis factor-α (TNF-α) were analysis by Porcine ELISA Kit (Beyotime Biotechnology Co., Ltd., Shanghai, China). Secretory Immunoglobulin A (sIgA) contents in the jejunum and colon were analyzed by Porcine ELISA Kit (TZIBIOTECH, Biotechnology Co., Ltd., Shanghai, China).

### 2.4. Histomorphology Analysis

The colon and jejunum were embedded in paraffin wax after being dried in graded alcohol and preserved with paraformaldehyde. Then, using a Zeiss Axio Imager microscope, bright-field images of paraffin slices that had been stained with hematoxylin and eosin (H&E) were taken. Images were evaluated using the Image J software (US National Institutes of Health, MD, USA), and at least 10 visual fields were selected for statistics. To determine the number of goblet cells in the colon and jejunum, PAS staining is carried out as directed. After routine dewaxing, add 3% acetic acid and incubate at room temperature for 3 min. Then, remove the acetic acid, add the PAS solution and incubate at 37 °C for 15 min. Rinse with 3% acetic acid solution for 10 s. Wash several times with distilled water for 4 min. Finally, dehydrate and seal the tablets. Let them dry overnight.

### 2.5. Short Chain Fatty Acids (SCFAs) Analysis

Piglet jejunal and colonic digesta SCFA concentrations were examined using a high-performance ion chromatography system (DIONEX ICS-3000; Thermo Fisher, Waltham, MA, USA). A 10 mL centrifuge tube was filled with 0.5 g of chyme content. For two tubes, each sample was weighed. To mix the contents, 8 mL of ultrapure water was added and vortexed. After 15 min of centrifugation at 4000 r/min, 0.16 mL supernatant was removed, and 7.84 mL of ultrapure water solution (equivalent to a 50-fold dilution) was added. The sample was then filtered, diluted with a 0.22 m mem-brane, and analyzed (Dionex IonPac AS11-HC; Thermo Fisher, Waltham, MA, USA).

### 2.6. Real-Time Quantitative PCR

Total RNA was extracted from ileal and colonic mucosa using an Ultra-pure Total RNA Extraction Kit (Aidlab, Biotechnology Co., Ltd., Beijing, China). All primers were synthesized by Beijing Qingke Biotechnology Co., Ltd. (Beijing, China), which are shown in Table 2. The mRNA expression of each gene was analyzed in triplicate on one plate per gene. Target gene mRNA abundance was normalized using the reference gene β-Actin and calculated using the 2−ΔΔCt method.

### 2.7. Microbial Analysis

Microbial DNA was extracted from segments using the QIAamp R Fast DNA Stool Mini Kit (Qiagen Ltd., Hilden, Germany) according to the manufacturer’s instructions. The V3-V4 regions of the bacteria 16S rRNA gene were generated with universal primers 338F (5′-ACTCCTACGGGAGGCAGCAG-3′) and 806R (5′-GGACTACHVGGGTWTCTAAT-3′) by PCR amplification [14]. Shanghai Majorbio Bio-pharm Technology Co. did the sequencing, Ltd. Purified amplicons were pooled in equimolar proportions and sequenced paired-end on the Illumina MiSeq platform (Illumina, San Diego, IL, USA) [15]. Sequence noise reduction using DADA2, species annotation database as silva138/16s_bacteria, species annotation method as bayes. A Naive Bayes classifier trained on the Greengenes 13 8 99% OTUs was used for taxonomic analysis. The raw sequencing data were processed by a previous study [14]. R was used to conduct statistical analyses (version 3.3.1). The vegan package was used to compute the Shannon index and richness (mothur-1.30). The data were analyzed on the online platform of Majorbio Cloud Platform (www.majorbio.com (accesed on 21 July 2022). All the raw data were uploaded into the NCBI Sequence Read Archive database with accession number PRJNA876600.

### 2.8. Statistical Analysis

The raw data were analyzed using the GLM model in SAS 9.2 statistical software (NC, USA), with each replicate as a statistical unit. The various dietary treatments had fixed effects, while the animal status and block had random effects. The UNIVARIATE procedure was used to look for outliers in the model residuals and the normal distribution with equal variance. For multiple comparisons, Tukey’s test was used, and the results were expressed as least squares means and mean standard errors. When *p* < 0.05, it is considered a significant difference, and when 0.05 < *p* < 0.10, it is considered a trend toward differences.

## 3. Results

### 3.1. Growth Performance and Diarrhea Incidence

The results for the effects of joint application of *Lactobacillus plantarum* and *Bacillus subtilis* on growth performance and diarrhea incidence of weaned piglets were shown in Table 3. There were no significant differences in the body weight of piglets among all dietary treatments. However, compared with the NC and PC groups, the LT group increased the body weight of weaned piglets by 2.32 kg and 1.12 kg on d 42, respectively. The diets containing 1 kg/t, 1.5 kg/t and 2 kg/t *Lactobacillus plantarum* and *Bacillus subtilis* significantly increased ADFI and ADG of weaned piglets in d 14–28 and d 28–42 compared with the NC group (*p* < 0.05). Groups of MT and HT reduced the FCR of weaned piglets in d 14–28 (*p* < 0.05). The LT diet increased the ADFI and ADG of weaned piglets in d 14–28 and d 28–42 compared with the PC group (*p* < 0.05). There was no significant effect on the incidence of diarrhea in weaned piglets among all the dietary groups.

### 3.2. Liver Function and Intestinal Permeability

The results for the effects of joint application of *Lactobacillus plantarum* and *Bacillus subtilis* on liver function and intestinal permeability of weaned piglets are shown in Figure 1. The results showed no significant differences in the serum AST, ALT, ALP, and D-lactic acid in piglets among all dietary treatments (Figure 1a,c). Compared with the NC group, the diets containing 1 kg/t, 1.5 kg/t, and 2 kg/t *Lactobacillus plantarum* and *Bacillus subtilis* reduced serum DAO concentration (*p* < 0.05) (Figure 1b). And groups of 1.5 kg/t and 2 kg/t significantly reduced serum DAO concentration compared with the PC group (*p* < 0.05).

### 3.3. Serum Antioxidant Capacity and Immune Function

The results for the effects of joint application of *Lactobacillus plantarum* and *Bacillus subtilis* on serum antioxidant capacity and immune functions in weaned piglets are shown in Figure 2. The results showed no significant differences in serum concentrations of IgM, IgA, IgG, SOD, MDA, T-AOC and IL-10 in weaned piglets among all dietary treatments (Figure 2a–c). The diets containing 1 kg/t, 1.5 kg/t and 2 kg/t *Lactobacillus plantarum* and *Bacillus subtilis* reduced serum concentrations of IL-1β and IL-2 compared with the PC group (*p* < 0.05), while the group of MT reduced serum concentration of TNF-α compared with the PC group (*p* < 0.05) (Figure 2c).

### 3.4. Intestinal Antioxidant Capacity and Immune Function

The results for the effects of joint application of *Lactobacillus plantarum* and *Bacillus subtilis* on intestinal antioxidant capacity and immune functions in weaned piglets are shown in Figure 3. The results showed that a group of 1 kg/t increased jejunal and colonic mucosal sIgA contents compared with NC and PC groups (*p* < 0.05) (Figure 3a,d). The diets containing 1 kg/t, 1.5 kg/t and 2 kg/t *Lactobacillus plantarum* and *Bacillus subtilis* decreased the concentration of IL-2 in the jejunal and colonic mucosa compared to the NC group (*p* < 0.05) (Figure 3b,e), while groups of 1 kg/t and 2 kg/t significantly decreased a concentration of IL-1β in the colonic mucosa. A group of 1 kg/t increased the concentration of IL-10 in the jejunal mucosa compared with the PC group (Figure 3b). There was no significant difference in the jejunal activity of SOD or T-AOC and a concentration of MDA (Figure 3c). However, supplementation of 1 kg/t, 1.5 kg/t and 2 kg/t *Lactobacillus plantarum* and *Bacillus subtilis* reduced MDA content in the colonic mucosa of piglets compared with the NC group (*p* < 0.05) (Figure 3f). Compared with the PC group, a group of 1 kg/t significantly increased colonic mucosal SOD activity in piglets compared with the PC group (*p* < 0.05). And the diets containing 1 kg/t, 1.5 kg/t, and 2 kg/t *Lactobacillus plantarum* and *Bacillus subtilis* increased the activity of SOD in the colonic mucosa compared with the NC group (*p* < 0.05).

### 3.5. Intestinal Morphology and Number of Goblet Cells

The results for the effects of joint application of *Lactobacillus plantarum* and *Bacillus subtilis* on intestinal morphology and the number of goblet cells in weaned piglets are shown in Figure 4. The results showed no significant differences in the jejunum and colonic crypt depth and numbers of goblet cells in the jejunum of weaned piglets among all dietary treatments. However, a group of 2 kg/t significantly increased the ratio of villus height to crypt depth (VH/CD) in the jejunum compared with the NC and PC groups (*p* < 0.05). Compared with the NC group, the number of colonic goblet cells in the colon of weaned piglets significantly increased (*p* < 0.05) in the 1 kg/t and PC groups compared with an NC group.

### 3.6. Lactic Acid and SCFA Concentrations

The results for the effects of joint application of *Lactobacillus plantarum* and *Bacillus subtilis* on intestinal lactic acid and SCFAs concentrations in weaned piglets are shown in Figure 5. The results showed no significant differences in jejunal acetic acid concentration and colonic propionic acid and butyric acid concentrations among all dietary treatments. A group of 2 kg/t significantly increased concentrations of jejunal lactic acid and total SCFAs in piglets compared with the NC group (*p* < 0.05).

### 3.7. Colonic Expression of Tight Junction Functions and Host Defense Peptides

The results for the effects of joint application of *Lactobacillus plantarum* and *Bacillus subtilis* on mRNA expression of tight junction functions and host defense peptides are shown in Figure 6. There was no significant difference in mRNA expression of colonic pBD-2 in weaned piglets (Figure 6a). In comparison with the NC and PC groups, adding 1.5 kg/t and 2 kg/t *Lactobacillus plantarum* and *Bacillus subtilis* to the diet remarkably increased mRNA expression of colonic pBD-1, pG1–5, Claudin-1, occluding, ZO-1 and mucin-1 in weaned piglets (*p* < 0.05) (Figure 6b).

### 3.8. Microbial Diversity

Effects of joint application of *Lactobacillus plantarum* and *Bacillus subtilis* on microbial α-diversity and community were shown in Appendix A. The results showed no significant differences in Shannon indexes of jejunal and colonic microbiota among all dietary treatments. In terms of gut microbial composition, a group increased a relative abundance of Proteobacteria at the phylum level in the jejunum (Appendix A), and the group increased a population of Lactobacillus in the jejunum at the level of the genus (Appendix A). There was a tendency (*p* < 0.10) for decreasing an abundance of genus Clostridium_sensu_stricto_1 in the diet containing 1 kg/t *Lactobacillus plantarum* and *Bacillus subtilis* compared with the NC group (Appendix A). Meanwhile, the group tended (*p* < 0.10) to increase the relative abundance of Lactobacillus in the colon compared with the NC and PC groups at the genus level (Figure 7).

## 4. Discussion

The positive effects of *Bacillus subtilis* or *Lactobacillus plantarum* alone or in combination on the growth performance of weaned piglets has been reported in previous studies [16,17,18]. In our study, joint application of *Bacillus subtilis* and *Lactobacillus plantarum* with an inclusion level of 1 kg/t in the diet improved the growth performance of weaned piglets d 14~28 and d 28~42 compared with the NC and PC groups. The positive effects on growth performance could be related to the improved intestinal integrity, immune function, and antioxidant capacity of weaned piglets after joint supplementation of *Bacillus subtilis* and *Lactobacillus plantarum.* However, joint application of *Bacillus subtilis* and *Lactobacillus plantarum* with different inclusion levels in the diet did not reduce diarrhea incidence of weaned piglets due to the inherently low diarrhea of piglets in our study. Serum D-lactate and DAO concentrations are always considered markers of intestinal permeability. D-lactate and DAO are absorbed into the blood by intestinal epithelial cells when intestinal integrity is damaged, resulting in greater concentrations of D-lactate and DAO [19]. The joint application of *Bacillus subtilis* and *Lactobacillus plantarum* with different inclusion levels in the diet reduced serum DAO concentration in the study, suggesting that joint application of *Bacillus subtilis* and *Lactobacillus plantarum* is beneficial to enhancing intestinal integrity in weaned piglets. The results for the reduced DAO content are consistent with previous studies in which dietary supplementation of *Bacillus subtilis* and *Lactobacillus plantarum* reduced serum DAO concentration in weaned piglets [8,20].

Intestinal villus height and density are critical indicators to reflect the nutrient absorption capacity of the host [21]. A previous study found that *Bacillus subtilis* increased the ratio of villus height to crypt depth in the jejunum of weaned piglets, and a joint addition of *Lactobacillus* and *Bacillus* increased the ratio of villus height to crypt depth in the ileum of pigs [8,22]. In this study, joint application of *Bacillus subtilis* and *Lactobacillus plantarum* at an inclusion level of 1 kg/t in the diet increased the ratio of villus height to crypt depth of piglets, which indicated that joint application of *Bacillus subtilis* and *Lactobacillus plantarum* helped to promote absorption of dietary nutrient in weaned piglets.

The piglet at weaning has an immature gastrointestinal tract and immune system, resulting in low abilities of anti-infection and nutrient digestion [23]. Piglets are exposed to oxidation stress at weaning, leading to a severe inflammatory response with increased pro-inflammatory cytokines concentrations and decreased levels of anti-inflammatory cytokines [1]. Compared with the PC group, joint application of *Bacillus subtilis* and *Lactobacillus plantarum* with different inclusion levels in the diet reduced concentrations of pro-inflammatory cytokines *IL-2* and *IL-1β* in the serum and intestinal mucosa of weaned piglets, which indicated that joint application of *Bacillus subtilis* and *Lactobacillus plantarum* could alleviate inflammatory responses caused by the oxidation stress of weaning. Supplementation of *Lactobacillus plantarum* and *Saccharomyces cerevisiae* in the sow diet increased the serum concentration of *IL-2* in piglets [24]. A previous study reported that *Lactobacillus plantarum* had an anti-inflammatory effect of improving the immune function of weaned piglets by reducing the expression of pro-inflammatory cytokines *IL-1β* and *IL-6* [25]. *Bacillus subtilis* also benefit to intestinal health of IUGR piglets by decreasing the expression of *IL-1β* in the intestine [12]. In addition, compared with the NC group, joint application of *Bacillus subtilis* and *Lactobacillus plantarum* with an inclusion level of 1 kg/t increased the activity of SOD and decreased MDA content in the colonic mucosa of weaned piglets, which suggested that joint application of *Bacillus subtilis* and *Lactobacillus plantarum* could improve the antioxidant capacity of weaned piglets. Maternal *Lactobacillus plantarum* and *Saccharomyces cerevisiae* supplementation can also increase the activity of SOD in Bama Mini-piglets [26]. A previous study showed that *Bacillus subtilis* could inhibit the production of reactive oxygen metabolites, such as MDA [27]. Furthermore, *Lactobacillus plantarum* exhibits antioxidant and cytoprotective capacity via increasing antioxidase activities in hydrogen peroxide-stimulated porcine intestinal epithelial cells [28].

Intestinal sIgA and host defense peptides are essential indicators of intestinal immune function in weaned piglets, which have anti-infectious and antimicrobial effects [29,30]. In our study, joint application of *Bacillus subtilis* and *Lactobacillus plantarum* with an inclusion level of 1 kg/t increased the sIgA content in the jejunal and colonic mucosal of weaned piglets. Meanwhile, joint application of *Bacillus subtilis* and *Lactobacillus plantarum* with an inclusion level of 1 kg/t and 2 kg/t increased colonic mRNA expression of pBD-1 and PG1-5 in piglets, which indicated that joint application of *Bacillus subtilis* and *Lactobacillus plantarum* improved immune function of weaned piglets. *Lactobacillus plantarum* isolated from the pig’s s intestine was reported to increase the synthesis of endogenous HDP by activating the TLR2/MAPK/AP-1 signaling pathway [9]. Another previous study showed *Lactobacillus plantarum* improves the immune function of the host by inhibiting toxin-producing *Escherichia coli* in pigs [31].

Tight junction proteins and mucins are composed of the intestinal integrity of the host, reflecting the function of the intestinal barrier. And *ZO-1*, *occludin* and *claudin-1* are essential structural and functional components of tight junction proteins [32,33]. The joint application of *Bacillus subtilis* and *Lactobacillus plantarum* with inclusion levels of 1.5 kg/t and 2 kg/t increased mRNA expression of colonic *claudin-1*, *occluding*, and *ZO-1* in weaned piglets compared with the NC and PC groups. It indicates that the Joint application of *Bacillus subtilis* and *Lactobacillus plantarum* improved the intestinal integrity of weaned piglets. In agreement with our finding, a previous study found that adding *Bacillus subtilis* to the diet increased the expression of *ZO-1* and *claudin-1* in the ileum of piglets [12].

In recent years, more and more studies have reported the relationship between gut microbial composition and host health [34,35]. However, joint application of *Bacillus subtilis* and *Lactobacillus plantarum* with different inclusion levels did not affect intestinal microbial diversity and composition in weaned piglets in our study, which indicated joint application of *Bacillus subtilis* and *Lactobacillus plantarum* had no obvious influences on the structure of gut microbiota. Inconsistent with our findings, a previous study reported that dietary supplementation with *Bacillus subtilis* increased the abundance of *Lactobacillus* in the intestine of weaned piglets [17]. In addition, lactic acid and SCFAs are the primary metabolites of the intestinal microbiota of dietary fiber components. Lactic acid can decrease intestinal pH to suppress the growth of pathogens and act on G protein-coupled receptor *Gpr81* to promote the development and regeneration of intestinal epithelial cell [36]. The SCFAs are an energy source for intestinal epithelial cells and play an essential role in the anti-inflammation of the host [37]. In our study, joint application of *Bacillus subtilis* and *Lactobacillus plantarum* at an inclusion level of 2 kg/t increased the concentration of lactic acid in the jejunal digesta of piglets compared with the NC group. However, joint application of *Bacillus subtilis* and *Lactobacillus plantarum* with different inclusion levels did not affect concentrations of acetic acid, propionic acid and butyric acid in the jejunal and colonic digesta. Therefore, the role of the joint application of *Bacillus subtilis* or *Lactobacillus plantarum* in regulating intestinal integrity by promoting intestinal microbial metabolism is still vague.

## 5. Conclusions

In conclusion, joint application of *Lactobacillus plantarum* and *Bacillus subtilis* improved growth performance, mRNA expression of tight junction protein and host defense peptides, and intestinal sIgA content in weaned piglets, as well as reducing the secretion of pro-inflammatory cytokines. Our finding povided new insight into improving growth performance, intestinal integrity and immune function via joint application of *Lactobacillus plantarum* and *Bacillus subtilis* as the antibiotics alternative, resulting in the improved growth performance of weaned piglets.

## Figures and Tables

**Figure 1 vetsci-09-00668-f001:**
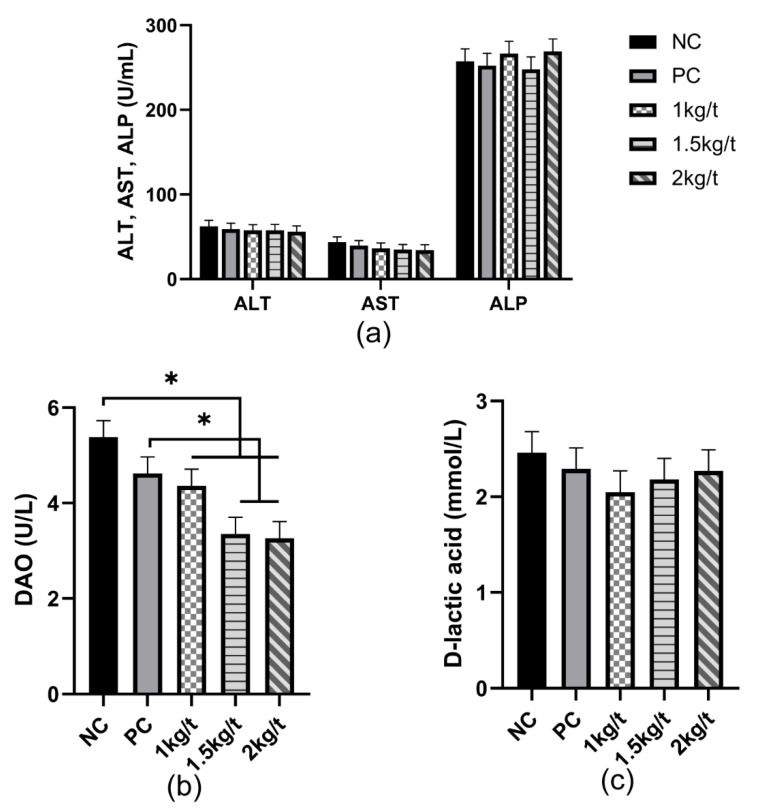
Effects of joint application of *Lactobacillus plantarum* and *Bacillus subtilis* on liver function and intestinal permeability in weaned piglets. (**a**) The amount of liver function-related indexes ALT, AST, and ALP in the serum of weaned piglets. (**b**,**c**) The amount of D-lactate and DAO, indicators related to intestinal permeability, in the serum of weaned piglets. “*” means there were statically significant differences (*p* < 0.05). NC, a control diet; PC, NC + 150 ppm mucilage sulfate.

**Figure 2 vetsci-09-00668-f002:**
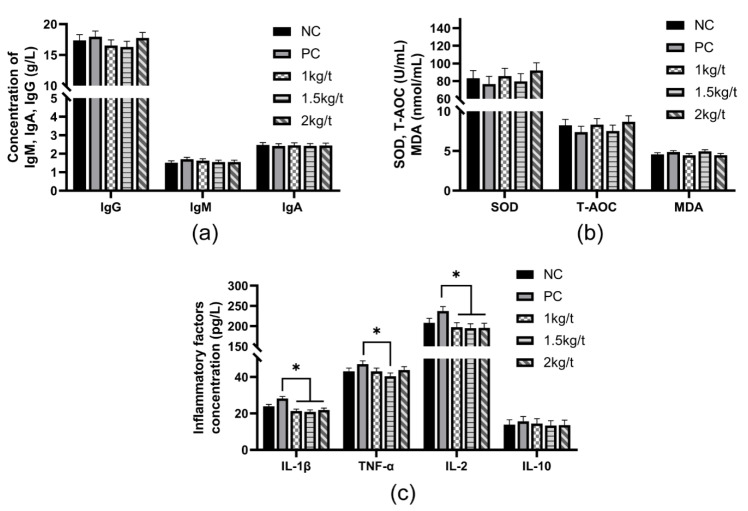
Effects of joint application of *Lactobacillus plantarum* and *Bacillus subtilis* on serum antioxidant capacity and immune function in weaned piglets. (**a**) The serum concentration of immunoglobulin in weaned piglets. (**b**) Serum concentrations of antioxidant-related substances in weaned piglets. (**c**) The serum concentration of inflammatory factors in weaned piglets. “*” means there were statically significant differences (*p* < 0.05). NC, a control diet; PC, NC + 150 ppm mucilage sulfate.

**Figure 3 vetsci-09-00668-f003:**
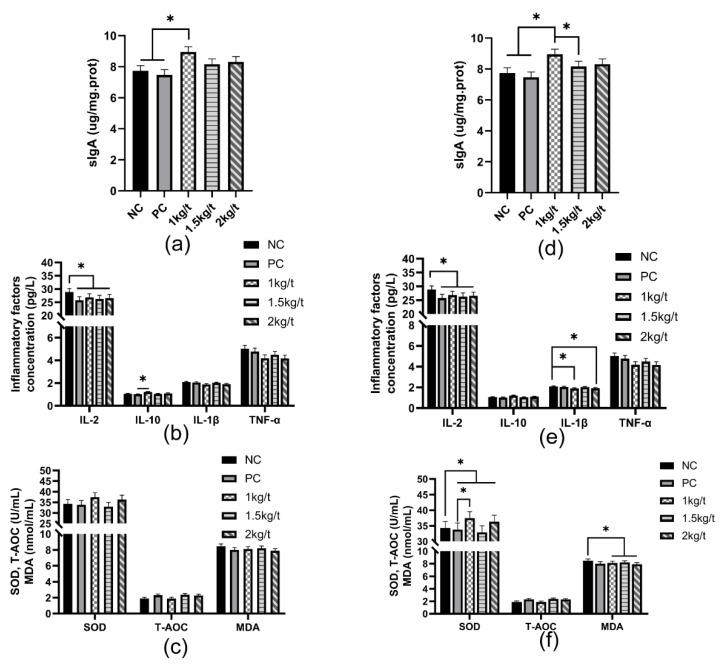
Effects of joint application of *Lactobacillus plantarum* and *Bacillus subtilis* on intestinal antioxidant capacity and immune function in weaned piglets. (**a**) Jejunum concentrations of sIgA in weaned piglets. (**b**) Jejunum concentrations of inflammatory factors in weaned piglets. (**c**) Jejunum concentrations of antioxidant-related substances in weaned piglets. (**d**–**f**) are the concentration of sIgA, inflammatory factors, and antioxidant-related substances in weaned piglets’ colons. “*” means there were statically significant differences (*p* < 0.05). NC, a control diet; PC, NC + 150 ppm mucilage sulfate.

**Figure 4 vetsci-09-00668-f004:**
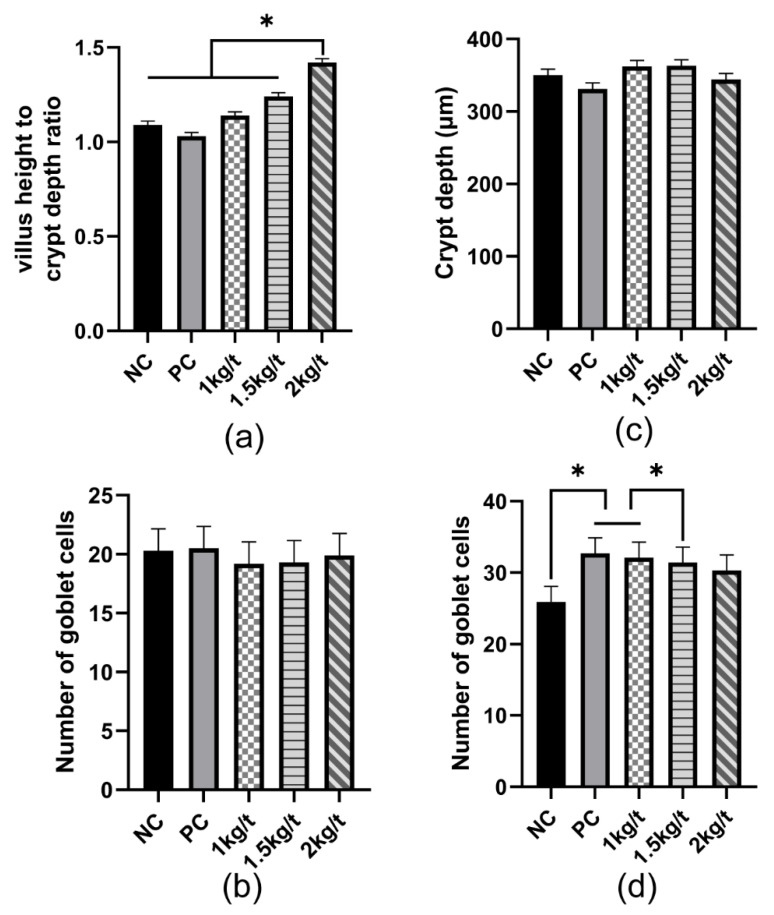
Effects of joint application of *Lactobacillus plantarum* and *Bacillus subtilis* on intestinal morphology and number of goblet cells in weaned piglets. (**a**) The ratio of villus height to crypt depth in the jejunum. (**b**) The number of jejunal goblet cells. (**c**) The depth of the colonic crypt. (**d**) The number of colonic goblet cells. “*” means there were statically significant differences (*p* < 0.05). NC, a control diet; PC, NC + 150 ppm mucilage sulfate.

**Figure 5 vetsci-09-00668-f005:**
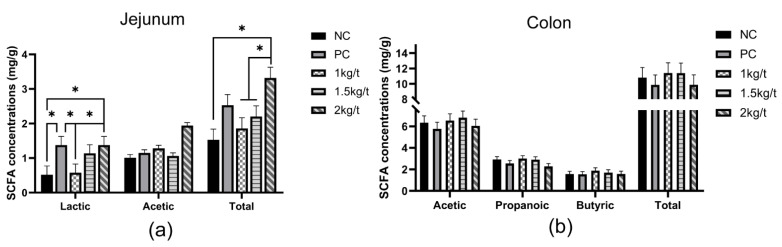
Effects of joint application of *Lactobacillus plantarum* and *Bacillus subtilis* on intestinal lactic acid and SCFAs concentrations in weaned piglets. (**a**) Lactic acid and SCFA content in the jejunal digesta. (**b**) SCFAs content in the colonic digesta. “*” means there were statically significant differences (*p* < 0.05). NC, a control diet; PC, NC + 150 ppm mucilage sulfate. Note: Propionic acid and butyric acid were not detected in jejunal digesta, and lactic acid was not detected in colonic digesta.

**Figure 6 vetsci-09-00668-f006:**
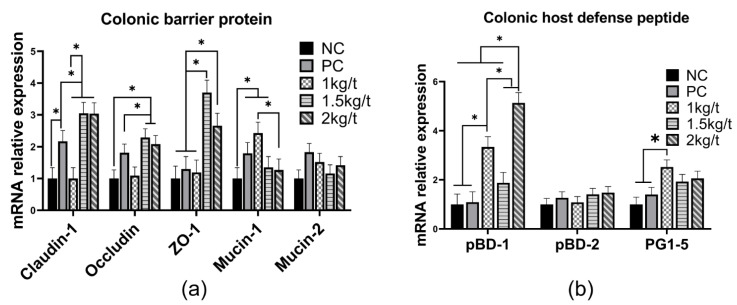
Effects of joint application of *Lactobacillus plantarum* and *Bacillus subtilis* on mRNA expression of colonic tight junction proteins protein and host defense peptides in weaned piglets. (**a**) The mRNA relative expression of colonic barrier protein. (**b**) The mRNA relative expression of colonic host defense peptide. “*” means there were statically significant differences (*p* < 0.05). NC, a control diet; PC, NC + 150 ppm mucilage sulfate.

**Figure 7 vetsci-09-00668-f007:**
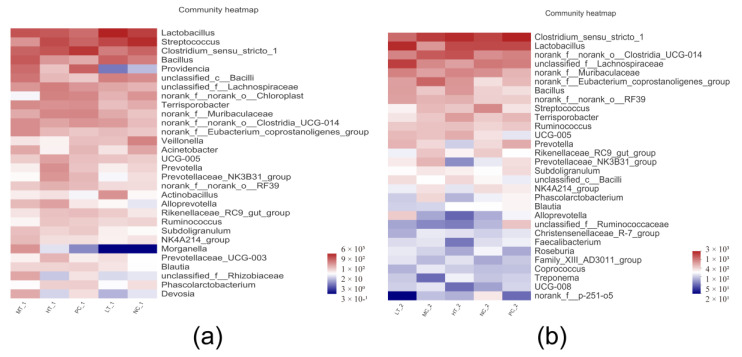
Heatmap in the jejunum and colonic digesta of weaned pigs at the genus level. (**a**) Community heatmap of jejunal microbial communities at the genus level. (**b**) Community heatmap of colonic microbial communities at the genus level. NC, a control diet; PC, NC + 150 ppm mucilage sulfate; LT: 1 kg/t joint application of *Lactobacillus plantarum* and *Bacillus subtilis*; MT: 1.5 kg/t joint application of *Lactobacillus plantarum* and *Bacillus subtilis*; HT: 2 kg/t joint application of *Lactobacillus plantarum* and *Bacillus subtilis*.

**Table 1 vetsci-09-00668-t001:** Ingredients composition and nutritive levels of the control diet (%, as-fed basis).

Ingredients, %	Control
Corn	56.45
Soybean meal	13.50
Whey powder	10.00
Fish meal	3.00
Soy protein concentrate	5.00
Extruded full-fat soybean	5.00
Sucrose	2.00
Soybean oil	1.30
Dicalcium phosphate	1.20
Limestone	0.75
NaCl	0.20
L-Lysine-HCl	0.45
DL-Methionine	0.20
L-Threonine	0.15
L-Tryptophan	0.10
L-Valine	0.20
Premix ^1^	0.50
Nutritive levels, %	
Digestible energy, MJ/kg ^2^	14.82
Crude protein	18.95
Calcium	0.83
Phosphorus	0.66

^1^ Premix provides: vitamin A, 12,000 IU; vitamin D3, 2500 IU; vitamin E, 30 IU; vitamin K3, 3 mg; vitamin B1, 2.5 mg; vitamin B2, 4.0 mg; vitamin B6, 3.0 mg; vitamin B12, 12 μg; niacin, 40 mg; thiamin, 3 mg; riboflavin, 6 mg; pantothenic acid, l5 mg; folic acid, 0.7 mg; biotin, 50 μg; iron, 90.0 mg; copper, 25.0 mg; zinc, 75.0 mg; manganese, 40 mg; iodine, 0.35 mg; selenium, 0.3 mg. ^2^ Calculated values.

**Table 2 vetsci-09-00668-t002:** Primer sequences for real-time PCR.

Target Genes	Primer	Primer Sequence (5′→3′)
β-Actin	Forward	CCAGGTCATCACCATCGGCAAC
	Reverse	CAGCACCGTGTTGGCGTAGAG
Mucin-1	Forward	GTGCCGACGAAAGAACTG
	Reverse	TGCCAGGTTCGAGTAAGAG
Mucin-2	Forward	CTGTGTGGGGCCTGACAA
	Reverse	AGTGCTTGCAGTCGAACTCA
ZO-1	Forward	GCCATCCACTCCTGCCTAT
	Reverse	CGGGACCTGCTCATAACTTC
Occludin	Forward	CAGCAGCAGTGGTAACTTGG
	Reverse	CAGCAGCAGTGGTAACTTGG
Claudin-1	Forward	AAGGACAAAACCGTGTGGGA
	Reverse	CTCTCCCCACATTCGAGATGATT
pBD-1	Forward	TGCCACAGGTGCCGATCT
	Reverse	CTGTTAGCTGCTTAAGGAATAAAGGC
pBD-2	Forward	CCAGAGGTCCGACCACTACA
	Reverse	GGTCCCTTCAATCCTGTTGAA
PG1-5	Forward	GTAGGTTCTGCGTCTGTGTCG
	Reverse	CAAATCCTTCACCGTCTACCA

**Table 3 vetsci-09-00668-t003:** Effects of joint application of *Lactobacillus plantarum* and *Bacillus subtilis* on growth performance and diarrhea rate of weaned piglets.

			Mixture of Probiotics				
Items	NC	PC	LT	MT	HT	SEM	*p*-Value
Body weight							
d 0	7.90	7.91	7.91	7.90	7.90	0.13	0.99
d 14	11.88	12.12	12.17	11.98	11.99	0.32	0.91
d 28	17.88	18.60	19.28	18.96	19.00	0.58	0.16
d 42	27.50	28.70	29.82	29.28	29.34	0.92	0.11
d 0–14							
ADFI	418	427	440	411	434	15	0.75
ADG	285	303	307	293	294	9	0.45
FCR	1.47	1.41	1.45	1.41	1.48	0.01	0.41
d 14–28							
ADFI	720 ^c^	743 ^bc^	812 ^a^	762 ^ab^	777 ^ab^	17	0.01
ADG	430 ^c^	465 ^b^	511 ^a^	500 ^a^	502 ^a^	11	0.01
FCR	1.68 ^a^	1.6 ^ab^	1.59 ^ab^	1.53 ^b^	1.55 ^b^	0.01	0.01
d 28–42							
ADFI	1245 ^c^	1287 ^bc^	1412 ^a^	1322 ^b^	1352 ^ab^	18	0.01
ADG	692 ^c^	710 ^bc^	758 ^a^	741 ^ab^	743 ^ab^	12	0.01
FCR	1.81	1.82	1.87	1.79	1.82	0.02	0.23
Diarrhea incidence, %							
d 0–14	1.72	1.06	2.51	0.93	1.46	0.83	0.89
d 14–28	3.57	1.98	2.65	1.72	3.44	1.04	0.74
d 0–28	2.79	1.57	2.72	1.35	2.60	0.94	0.81

^a,b,c^ Different letters mean there were statistically significant differences among the three treatments when *p*-value < 0.05. ADFI, Average daily feed intake; ADG, average daily gain; FCR, feed conversion ratio; NC, a control diet; PC, NC + 150 ppm mucilage sulfate; LT: 1 kg/t joint application of *Lactobacillus plantarum* and *Bacillus subtilis*; MT: 1.5 kg/t joint application of *Lactobacillus plantarum* and *Bacillus subtilis*; HT: 2 kg/t Joint application of *Lactobacillus plantarum* and *Bacillus subtilis*.

## Data Availability

All the raw data about bacteria 16S rRNA gene were uploaded into the NCBI Sequence Read Archive database with accession number PRJNA876600.

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
