# Peer review of "Joint Application of Lactobacillus plantarum and Bacillus subtilis Improves Growth Performance, Immune Function and Intestinal Integrity in Weaned Piglets"

_vetsci, 2022, doi:10.3390/vetsci9120668_

Round 1

Reviewer 1 Report

The authors have tried to establish a mixture of two different probiotics as a potential alternative for antibiotics. No data are given for the exact dosage based on CFU of both probiotics are given. The approach to measure feed intake is not clear. From the huge number of different parameters it is not possible to identify a clear hypothesis for their work

Author Response

Response to Manuscript vetsci-1962359 entitled “Joint application of Lactobacillus Plantarum and Bacillus subtilis improves growth performance, immune function and intestinal integrity in weaned piglets”.

Dear Editors and Reviewers,

We are very glad to receive your letter regarding our manuscript. We have carefully considered your constructive comments and have revised our manuscript accordingly. We also try our best to reduce similarity of the manuscript. All changes made to the manuscript are highlighted. We would like to thank you for your support of our work as well as the reviewers for their helpful comments to improve our manuscript. We hope that the revised manuscript is now acceptable for publication. We are looking forward to your feedback.

Yours sincerely,

Jinbiao Zhao,

State Key Laboratory of Animal Nutrition, Ministry of Agriculture Feed Industry Centre, China Agricultural University

Response to Reviewer 1 Comments

Point 1: The authors have tried to establish a mixture of two different probiotics as a potential alternative for antibiotics. No data are given for the exact dosage based on CFU of both probiotics are given. The approach to measure feed intake is not clear. From the huge number of different parameters it is not possible to identify a clear hypothesis for their work.

Response 1: Thanks for your comments. Firstly, we provided the exact inclusion dosage of both probiotics to the diets in the original manuscript. A mixture of probiotics is composed of Lactobacillus Plantarum (2.0×108 CFU/g) and Bacillus subtilis (1.3×109 CFU/g), and the inclusion dosages of probiotics are 1 kg/t, 1.5 kg/t and 2 kg/t, respectively. Please see Line 77-79 and Line 85-87.

Secondly, we have supplemented more details in evaluating growth performance of piglets. In the feeding trial, all piglets were weighed individually on day 14, 28, and 42 to calculate average daily gain (ADG). The piglets' feed consumption in each pen was recorded from d 1-14, d 14-28 and d 28-42 to calculate average daily feed intake (ADFI). Feed conversion ratio (FRC) was defined as a ration of ADFI to ADG. Please see Line 103-106.

Reviewer 2 Report

The authors studied explore the effects of the joint application of Lactobacillus Plantarum and Bacillus subtilis on growth performance, immune function, antioxidant capacity, the intestinal integrity and composition of the gut microbiota in weaning.
The authors combined the use of different techniques: molecular and quantitative histological to obtain  your results.
The manuscript is well written, the data is well presented and discussed. I would recommend this publication as is.

Author Response

Response to Manuscript vetsci-1962359 entitled “Joint application of Lactobacillus Plantarum and Bacillus subtilis improves growth performance, immune function and intestinal integrity in weaned piglets”.

Dear Editors and Reviewers,

We are very glad to receive your letter regarding our manuscript. We have carefully considered your constructive comments and have revised our manuscript accordingly. We also try our best to reduce similarity of the manuscript. All changes made to the manuscript are highlighted. We would like to thank you for your support of our work as well as the reviewers for their helpful comments to improve our manuscript. We hope that the revised manuscript is now acceptable for publication. We are looking forward to your feedback.

Yours sincerely,

Jinbiao Zhao,

State Key Laboratory of Animal Nutrition, Ministry of Agriculture Feed Industry Centre, China Agricultural University

Response to Reviewer 2 Comments

Point 1: The authors studied explore the effects of the joint application of Lactobacillus Plantarum and Bacillus subtilis on growth performance, immune function, antioxidant capacity, the intestinal integrity and composition of the gut microbiota in weaning.

The authors combined the use of different techniques: molecular and quantitative histological to obtain your results.

The manuscript is well written, the data is well presented and discussed. I would recommend this publication as is.

Response 1: Thanks for your comments. We have further revised our manuscript to improve quality of the manuscript based on all comments and suggestions of the reviews. All revisions were highlighted in the red font.

Reviewer 3 Report

The publication addresses an interesting research topic. The authors give piglets a mixture of Lactobacillus Plantarum and Bacillus subtilis at different levels. It would be interesting to compare the results obtained with groups receiving only Lactobacillus Plantarum or only Bacillus subtilis. The authors compare the results obtained with literature data on where these probiotics were used, but they do not state in what amount or for how long. These factors affect the results obtained. Based on this experimental arrangement, it is hard to say whether these probiotics work better together or whether it is better to use one of them. 

Why precisely such amounts of Lactobacillus Plantarum (≥ 2.0×108 CFU/g) and Bacillus subtilis (≥ 1.3×109 CFU/g) in the mixture?

Why did the authors use such doses? This information should be justified. 

In addition, what is the purpose of using positive control? What is mucilage sulfate? This information should be supplemented. 

There are no labeled groups in the tables with results, while in the results section, the authors describe the results using group names. This is confusing. 

In Table 3, based on statistical analysis, it can be concluded that there is no effect of the experimental factor on body weight. Why does the description state that there is an increase in body weight when there is no statistical significance? Similarly, in the discussion, the authors state a performance improvement. Can such a statement be made if there is no increase in piglet weights, but only changes in other parameters?

The gut results also seem puzzling. Why, if there is an improvement in intestinal performance, is there no effect on the incidence of diarrhea?

Author Response

Response to Manuscript vetsci-1962359 entitled “Joint application of Lactobacillus Plantarum and Bacillus subtilis improves growth performance, immune function and intestinal integrity in weaned piglets”.

Dear Editors and Reviewers,

We are very glad to receive your letter regarding our manuscript. We have carefully considered your constructive comments and have revised our manuscript accordingly. We also try our best to reduce similarity of the manuscript. All changes made to the manuscript are highlighted. We would like to thank you for your support of our work as well as the reviewers for their helpful comments to improve our manuscript. We hope that the revised manuscript is now acceptable for publication. We are looking forward to your feedback.

Yours sincerely,

Jinbiao Zhao,

State Key Laboratory of Animal Nutrition, Ministry of Agriculture Feed Industry Centre, China Agricultural University

Response to Reviewer 3 Comments

Point 1: Why precisely such amounts of Lactobacillus Plantarum (≥ 2.0×108 CFU/g) and Bacillus subtilis (≥ 1.3×109 CFU/g) in the mixture? Why did the authors use such doses? This information should be justified.

Response 1: Previous data in practice indicate that it is when numbers of 1.0×108 CFU/g Lactobacillus Plantarum or 1×109 CFU/g Bacillus subtilis could imporve weaned piglets’ performance. Then, a mixture of Lactobacillus Plantarum and Bacillus subtilis were cultured in vitro fermentation. After optimizing culture conditions, the mixture of 2.0×108 CFU/g Lactobacillus Plantarum and 1.3×109 CFU/g Bacillus subtilis was produced and used in our study. Please see Line 74-79.

Point 2: In addition, what is the purpose of using positive control? What is mucilage sulfate? This information should be supplemented.

Response 2: The positive control diet was chosen to use mucilage sulfate, which is one of the antibiotics used in the past for growth promotion in animal husbandry. Now, supplementation of the antibiotics has been prohibited due to production of drug resistant genes. In order to compare whether the addition of probiotics have a growth-promoting effect, and to illustrate the effect of adding a complex of probiotics on growth performance compared with the antibiotics. This information has been supplemented in section 2.1. See Line 87-90.

Point 3: There are no labeled groups in the tables with results, while in the results section, the authors describe the results using group names. This is confusing.

Response 3: Changes have been made in the result tables and images where the group names do not correspond to the descriptions of the results in the corresponding locations. As the description in section 2.1: Dietary treatments included a control diet (NC), NC + 150 ppm mucilage sulfate (PC) and 3 additional diets containing 1 kg/t (LT), 1.5 kg/t (MT), 2 kg/t (HT) mixture of Lactobacillus Plantarum (2.0×108 CFU/g) and Bacillus subtilis (1.3×109 CFU/g). See Line 85-87.

Point 4: In Table 3, based on statistical analysis, it can be concluded that there is no effect of the experimental factor on body weight. Why does the description state that there is an increase in body weight when there is no statistical significance? Similarly, in the discussion, the authors state a performance improvement. Can such a statement be made if there is no increase in piglet weights, but only changes in other parameters?

Response 4: In pig feeding trial, we describe the results of body weight based on the numerical differences. Although no significant differences in the body weight of piglets were observed in our study, but it is also meaningful when the diet including 1 kg/t a mixture of Lactobacillus Plantarum and Bacillus subtilis increased body weight of weaned piglets by 2.32 kg and 1.12 kg on d 42 in practice. To avoid the readers misunderstanding, we have revised the description as following “There were no significant differences in the body weight of piglets among all dietary treatments. However, compared with the NC and PC groups, the 1kg/t group increased body weight of weaned piglets by 2.32 kg and 1.12 kg on d 42, respectively. “ See Line 197-198.

 In addition, parameters of growth performance include ADFI, ADG and FCR, as well as pig body weight. In our study, the diets containing 1 kg/t, 1.5 kg/t and 2 kg/t Lactobacillus Plantarum and Bacillus subtilis significantly increased ADFI and ADG of weaned piglets in d 14-28 and d 28-42 compared with the NC group (P < 0.05). Groups of 1.5kg/t and 2kg/t reduced the FCR of weaned piglets in d 14-28 (P < 0.05). The 1kg/t diet increased the ADFI and ADG of weaned piglets in d 14-28 and d 28-42 compared with the PC group (P < 0.05). Therefore, the results mentioned above indicated Lactobacillus Plantarum and Bacillus subtilis can improve pig performance.

Point 5: The gut results also seem puzzling. Why, if there is an improvement in intestinal performance, is there no effect on the incidence of diarrhea?

Response 5: The piglets used in our study have a low rate of diarrhea, as mentioned in the first paragraph of section 4, Line 430~432: However, joint application of Bacillus subtilis and Lactobacillus Plantarum with different inclusion levels in the diet did not reduce diarrhea incidence of weaned piglets, which should be associated with the inherently low diarrhea of piglets in our study. And there were significant changes in some indicators related to improvement of gut health in the results. For example, there was a significant change in DAO, in the study, and DAO are absorbed into the blood by intestinal epithelial cells when intestinal integrity is damaged, resulting in greater concentrations of DAO. Also, there was a improvement in the ratio of villus height to crypt depth in the pigs fed a diet containing 2 kg/t a mixture of Lactobacillus Plantarum and Bacillus subtilis.

Round 2

Reviewer 3 Report

The authors corrected the manuscript according to my suggestions.

Author Response

Done.